# Fraternity as Natural Being

**Joachim Ostermann, OFM**

Franciscans of Canada, Montreal, QC H1T2H2, Canada; joachim.ostermann@gmail.com

**Abstract:** In a scientifically understood world, making sense of natural being is a challenge. This is particularly acute when knowledge of nature impinges on human autonomy. I present two examples: The legitimacy of opposing abortion on account of "potential life", and the legitimacy of mandatory vaccination during a pandemic. I then explore the concept of fraternity in the writings of St. Francis at the example of the Rule of 1221 and the Canticle of Creatures. In conclusion, I show how the concept of fraternity as applied in Franciscan life allows us to reconcile the relationship between natural being and human autonomy through relationships of mutual care.

**Keywords:** faith and science; nature; natural law





## 1. Introduction

"The Order of Friars Minor, founded by St. Francis of Assisi, is a fraternity" are the opening words of the Franciscan General Constitutions.[1] But what is meant by fraternity? What is the characteristic meaning and value of fraternity, and how does it define the understanding of ourselves and our relationships with each other? As I wish to show in this paper, fraternal life gives meaning and direction to the concept of human autonomy. At a time of increasing political polarization, understanding fraternity dispels the despair of futility and lets us persevere in the apparently Sisyphean task of healing lives damaged by political strife and environmental harm.

In one of St. Francis's best-known writings, the Canticle of the Creatures, the saint speaks of the celestial bodies, the sun, the moon, and the stars, and the classical elements, wind, water, fire, and earth, as brothers and sisters.[2] For him, fraternity is the concept by which he understands the relationships between all natural beings and all of God's creatures. All together, as one family, sing the praises of God. This raises a further question. What does it mean to think of non-human natural beings in fraternal relations with us and each other?

The modern age is the period in human history in which human life reached mastery over nature in ways unimaginable during the 13th century, which was the time when the Franciscan Order was founded, and the Canticle of the Creatures was composed. It is also the age of modern science and its triumph over two of the three traditional banes of human existence—pestilence and hunger—even if the third one, war, was held barely at bay by fear of mutually assured destruction.[3] Through the mastery of nature made possible by the scientific–technological revolution of modernity, the human family experienced unprecedented growth in the last 200 years.[4] But there are complex consequences from such an increase in power over nature.[5] Such an increase in power in human hands could easily lead to irreconcilable conflicts between different factions that have different conceptions of the good, of hopes for their future, and how these powers are to be used. A shared context is needed. Through much of humanity's history, human autonomy could only be understood as the freedom to act within the unbreakable bonds of nature. But understanding autonomy as rightful power over nature takes away the common ground of dependence on nature that lets different autonomous individuals turn towards a shared awareness of the good.

In this paper, I will consider nature as that what is presupposed in responsible human acting and within which it occurs. This does not mean drawing a line separating what is

natural from what is artificial. Mowing a lawn is human acting, and the lawn is part of nature in this act. At the same time, the cultivar of grass on the lawn that I mow is the result of human acting in selective breeding and growing under well controlled conditions. In the act of selecting lawn seed cultivars, what is natural is the wild grass from which the cultivar derived and its behavior during selective breeding and controlled growth. Defining nature in this way makes it a relative term to human acting. This definition comes closest to the way both ordinary language and modern science speak of nature. It also serves to protect from overly naïve naturalism. It does not ask whether a being or an activity is natural or artificial, but it asks in what sense it is natural or artificial relative to responsible human acting.

Should autonomy from nature be sought without understanding human autonomy's dependence on nature, it could go as far as removing a necessary check on human autonomy that protects the powerless from the powerful: The shared human nature that is the foundation of the shared human dignity equally present in everyone, irrespective of their capabilities and accomplishments. If human beings were masters over nature, then human nature would become ours to master as well, and it would be mastered by those who are powerful, giving much to fear for those who are not. Expanding human autonomy in distinction from nature, or even against nature, might well be part of the reasons for political polarization that now seems to emerge as a common theme in the societies of the "West", or the society in which the scientific–technological revolution with its unprecedented mastery over nature began.

I suggest that we consider fraternity to be the healing response, as it is the response that puts human autonomy and human knowledge into a shared context of living in interdependence with others. To make this argument, I will begin with two examples for political polarization in which knowledge of nature conflicts with human autonomy. Then I will show the meaning of fraternity by exploring the Franciscan tradition. In conclusion, I will show how the concept of fraternity can be applied to understand conflict and facilitate mutual understanding.

## 2. Two Contemporary Examples of the Conflict

### 2.1. Abortion and Rights

My first example is taken from current US-American politics: The US Supreme Court's ruling in Dobbs v. Jackson Women's Health Organization that denies that abortion is a right protected under the US constitution.[6]

In this ruling, the majority finds that the US constitution grants no right to an abortion either explicitly or implicitly. It is not disputed by anyone that this constitution does not explicitly grant such a right. However, the question is whether any of the recognized rights of the constitution, such as the right to equal protection under the law or the right to autonomy in personal decision making, must today be interpreted to include a right to an abortion. The broader legal question is when it is the case that traditional basic rights must now be interpreted in light of changed circumstances so that they now include what was previously excluded. The majority finds that this is not the case here. The presence of what is called "potential life" during pregnancy grounds the court's understanding that this ruling must be different from other rulings that expanded other rights, such as the right to same-sex marriage.[7] When two lives are at stake, then neither is autonomous in one's actions towards another, but the community is autonomous in decision making, even when this restricts the autonomy of individual members of this community. The existence of potential life creates a situation in which the state has the right to pass laws through the ordinary deliberations of a democratic society by which the various interests of the state's citizens are ordered. If they so choose, then these laws may restrict abortions.

The minority opinion argues that a right to abortion does exist and is protected from state interference, as it is an essential dimension of a woman's control over the integrity of her own body that cannot be lawfully rendered to the state. Prohibiting abortion is seen to violate one of the original rights of every human being that is not ceded to the state by the

acceptance of a constitution as a foundational political order. Not offering the possibility of an abortion to a pregnant woman is seen as an illegitimate interference with the woman's autonomy in decision making in her life that limits a woman's right to fully participate in the life of the community in any which way that she chooses.

Obviously, the potential life referred to by the majority is a concept that cannot be interpreted too broadly. It does not refer to children that are merely anticipated as a future possibility, but the human potential of the biologically distinct human being whose life has already begun when abortion is considered. This leaves open questions as to the relationship between legally protected human life and human life as biology, and the US Supreme Court does not attempt to answer this question. Biologically, there is no doubt that an embryo or fetus is a distinct human living being with its own developmental past and future.[8] At the same time, a bond may or may not be formed between the woman and the child-to-be-born. How the woman and others in her life perceive the pregnancy—either as her child-to-be-born or an unwanted interference in the pregnant woman's life with potentially catastrophic consequences—is a matter entirely outside of biology. At this time, this court leaves the decision to what extent unborn human life is protected to the political process of lawmaking.

It is instructive to look at Germany for insights from a different jurisdiction. There, abortion is conditionally tolerated. In 1993, the German supreme court reaffirmed in what is its most recent ruling on the matter that there cannot be a right to abortion under the German constitution.[9] It refers to the first article of the German constitution states that "human dignity shall be inviolable. To respect and protect it shall be the duty of all state authority".[10] The second article further states that "everyone shall have the right to life and physical integrity. Freedom of the person shall be inviolable. These rights may be interfered with only pursuant to a law".[11] The German supreme court currently takes the position that the state's obligation to protect human dignity and human life rules out that the state recognizes a right to abortion. However, German law also recognizes that early in the pregnancy, an abortion under certain conditions is not being sanctioned. In other words, it remains unlawful but is tolerated. In this way, a compromise was reached that respects the political will to make abortion an option for the pregnant woman under certain circumstances without denying the state's duty to protect human life.

*2.2. Vaccine Mandates*

The second example is mandatory vaccination. Such a mandate also gives rise to the question whether biological knowledge can be binding on ethical decision making. During the COVID-19 pandemic, vaccines were developed at an unprecedented speed. They limited the disease's impact on health care systems that were overburdened by the rapidly increasing number of patients requiring intensive medical care. The spread of this novel virus in the human population caused an evident public health requirement to vaccinate as many as possible as fast as possible. However, public pressure against vaccine mandates was so strong that vaccine mandates remained limited.

Vaccine mandates were not always considered problematic. Smallpox vaccination was mandatory for all until this disease was considered eradicated, even though the smallpox vaccine came with a considerable risk, while the COVID-19 vaccines are safer than ordinary life.[12] What is new is not the mandate to vaccinate on account of such a mandate being in the public interest, but the political acrimony with which the matter was discussed. Clearly, the public has become much less tolerant to government instructions regarding one's health. Personal autonomy is seen as so important that it trumps the government's mandate of protecting the public in a natural disaster even when objective evidence establishes both a substantial individual and a collective benefit. Personal autonomy is now even called upon by those advocating their own extra-institutional understanding of modern science and its results.[13] The authority of scientists to speak from a privileged point of view on account of their professional training and standing in the scientific community is now routinely questioned.

What emerges as the common theme between these two very different examples is the relationship between, on the one hand, personal autonomy in matters of ethical choice and, on the other hand, universal biology as understood by modern science. The universality of biology is juxtaposed with the individuality of autonomy with nothing in between. I chose these two examples to establish that the resulting political acrimony does not follow traditional political lines of division between progressives and conservatives. This is not about collective vs. individual decision making and political theories favoring one over the other. Personal autonomy can be a rallying cry for both conservatives and progressives. But here, the topic is not to study why either group chooses personal autonomy in one but not the other situation. Maybe it is simply a historical artifact of past political developments that had either camp crystalize around either position. When two true principles, such as the autonomy of the human person and the biology of a human being, are left juxtaposed while reason requires them to be reconciled, then unpredictable strife seems inevitable.

## 3. Understanding Fraternity

### 3.1. Biology and Ethics

Fraternity is a concept that from the first beginning of its understanding already intends both a biological and an ethical reality. At its simplest, brothers or sisters denote individuals who share the same parents and whose relationship is shaped by being children in the same family. They contribute to and share dependencies on the same pool of resources. They discover a sense of fairness in this relationship. As adults, they continue to share inherited features from their parents and childhood experiences that they carry into their adult life. Now independent of each other and their parents, they may retain a bond that allows one to rely on the other in times of need. They may consider this bond and its obligations to care involuntary, as a demand inherent in their nature. Furthermore, the bond of fraternity established in their interdependency can make bearable what is otherwise a dispiriting existential loneliness of personal being that seeks to be known by another. However, this fraternal bond is not merely a matter of biology and biological instinct. Non-related others may be adopted into the same kind of relationship. The recognition of brothers and sisters by adoption or some other way, such as through fraternal organizations, expands applications of the concept from its roots in biology to the larger context of the human ethical order. We can begin with biology, understand it, and then transcend biology's limited understanding by fulfilling it without losing our grounding in biology. Then, the meaning of fraternity can even become a universal concept embracing the whole human race, as it does in Article 1 of the Universal Declaration of Human Rights: "All human beings are born free and equal in dignity and rights. They are endowed with reason and conscience and should act towards one another in a spirit of brotherhood".[14] In this sentence, the transition from what human beings are and how they ought to act is made through the concept of fraternity.

But the genetic origin of the meaning of fraternity is the ethical experience of children growing up together and understanding each other in through their dependencies and shared caregivers. Expectations regarding proper interpersonal peer relationships are established in the experience of fraternal relationships. This makes fraternity a biological concept that can be seamlessly transformed into an ethical concept while retaining its univocal meaning: Being with another one like I am someone, being with someone whose needs and obligations are like my own, being with someone who needs to be sustained as I must be sustained. It is a concept of equality with me, but it is also a concept of distinction from me. The other and I are not one, but we are in relationship, and it is a relationship of dependencies on care by others. It is this grounding in the experience of dependency and care—even when it might be a competition for insufficient care in less idealized circumstances—that defines fraternity and makes it a suitable concept that can be placed in between autonomy and hierarchically ordered structures of authority and obedience.

### 3.2. Fraternity in Religious Life

While fraternity is a universal concept in all cultures and religions, I wish to explore it specifically in Franciscan life. The meaning of fraternity in Franciscan life is best seen when the Franciscan rule is considered against the background of the oldest rule for Christian religious life still in use today. This is the Rule of St. Benedict.[15] This comparison is not meant to suggest that fraternity is a Franciscan invention that distinguishes them from others. St. Benedict also speaks of the monks as brothers, but the concept of fraternity formed by them is not a significant theme in the rule itself. His prologue addresses the individual monk in his relationship with God, and his rule is meant to direct the individual monk's conduct towards a deeper spiritual life. Monasticism remains fundamentally defined by the solitary life of each monk even when it is lived as communal life by the Rule of St. Benedict. In as much as they are a community, the monks are like hermits who share a dwelling and cooperate in the upkeep of their collective hermitage. Therefore, silence is an important part of their life.[16]

Considering the Franciscan rule requires knowledge of its complex history. The rule that was officially approved is the rule of 1223, which follows the Church's Code of Canon Law with its expectations for a well-ordered religious community. For those wanting to understand the meaning of fraternity in the Franciscan tradition, more insightful is an earlier rule, now referred to as the Rule of 1221, which was never formally approved and appears to have been a community rule that was developed and amended as needed by the early Franciscan community (Flood and Matura 1975). The grounding of this rule in the history of the early Franciscan community gives us insights into their self-understanding as a fraternity, without being constrained by the ordinary expectations of the hierarchically ordered Roman Catholic Church for proper governance of a religious order.

Whereas the Rule of St. Benedict understands itself as "a school for the service of the Lord"[17], the 1221 Rule speaks of "following the teaching and footprints of our Lord Jesus Christ".[18] A school has teachers and students, but the Christian discipleship of which the Gospels speak is a community of equals that has only one master and teacher, Jesus Christ. Towards the end, in Chapter XXII, the 1221 Rule quotes Mt 23:8-10 ( . . . do not call yourself teachers . . . you have one teacher . . . you are all brothers and sisters . . . you have one Father, the one in heaven . . . ").[19] Whereas the Rule of St. Benedict is a rule for those few who wish to become perfect, the Franciscan rule responds more directly to the universal call to holiness that goes to all men and women.

Right after defining the Franciscan community in this way, the 1221 Rule speaks in the second chapter of the manner in which newcomers to the community are to be received. They are to be received with kindness, encouragement, and the Franciscan way of life is to be explained to them. Thereafter, they may enter the community for a year of probation. The Rule of St. Benedict does not address the receival of new members until the 58th chapter. The newcomer is to be tested by receiving him or her with harshness so that their ability to persevere is assessed. He or she needs to be suitable to bear the hard and rugged ways of monastic life. Here, the newcomer is not meant to experience the monastic community as a welcoming fraternity, but as a means of testing whether the newcomer is capable of following Christ as perfectly as necessary for monastic life.[20]

The elitist spirit of monastic life is also expressed in the way leadership is understood. St. Benedict reminds the abbot or abbess of a monastery that they are called by the name of Superior, and they are supposedly superior to the others in the devotion to Christ and imitation of him.[21] They are called to be fathers, to take the place of Christ in the monastery who represents our true Father in Heaven. They are "to govern their disciples through their . . . teaching" and encourage the good and discipline those of "harder hearts and ruder minds".

In the 1221 Rule, a different emphasis is made. The words "school", "teacher", or "disciple" are not used.[22] Chapter VI stipulates that no one is to be called Prior, but all are called lesser brother. This is preceded by giving each brother recourse to their minister if

they cannot observe this form of life, and the minister is called upon to provide for them as he would wish to be provided for. Earlier, in Chapter IV, the ministers are reminded to be servants of their community and that the care of their brothers is entrusted to them. The brothers are called to obedience, but even the all-important religious vow to obedience is qualified by the addition "in all matters concerning the well-being of their soul and which are not contrary to our way of life". Ultimately, the obedience is not to the minister but to "our way of life". The authority of leadership is qualified by the authority of the community. Neither the minister nor the individual brother is autonomous in this way of life, but their personal autonomy is exercised within the needs and collective subordination of the community and their understanding of the Gospel way of life.

In the 1221 Rule, the name "Father" always refers to the first person of the Trinity. Rather than using paternal images as is traditional in the Church when speaking of authority, St. Francis uses the image of motherhood. "Let each one love and care for his brother as a mother loves and cares for her son" is written in Chapter IX. This chapter is already on the topic of dependency on support from others, as it opens with the instructions for begging alms, and it continues with instructing the brothers to make their needs known to each other so that they can minister to each other. It recognizes the brothers' dependencies and needs that are met by the kindness of others. Motherhood is used as an image not only here but even more so in St. Francis's Rule for Hermitages.[23] There, brothers are to take turns being either mother or son, with the mother caring so that the son can spend his days in prayer. After a mutually agreed upon time, they change roles, and the son becomes the mother, and the mother becomes the son. Motherhood is used as an image rather than a category of lasting identity, but the initial contents of the image that defines its meaning is the actual experience of motherly care that many have received in infancy and becomes the idealized image of motherhood that Francis uses.

What emerges from this short reflection on fraternity is this. What the Rule of St. Benedict shares with the Franciscan 1221 Rule is that either rule is meant to lead the one living by this rule into a closer relationship with God. Either rule is meant to lead to eternal life. But where the Rule of St. Benedict places the superior into the mediating place between God and the brother or sister seeking God, the 1221 Rule places the community, and it is meant to be a community motivated by a spirit of maternal care. There is no suggestion that this emphasis on maternal care rather than paternal authority is just a preferred way of life, rather than an understanding of how to find closeness with God. The spirituality and the prayers written by St. Francis clearly are about the salvation of each individual soul. He seeks heaven, not worldly comforts, for himself and all others living like he does. But he sees the caring community and not an authoritarian superior as the means in which the rightful autonomy of the individual is disciplined and directed towards its proper goal—the Kingdom of God. This seems altogether safer, considering how hard it is to know everything about any one man and how often one is disappointed by deep flaws hidden behind even the most credible façade of holiness.[24] No one man's heart can ever be entirely unstained by evil—lest it is the heart of Jesus Christ. The mutuality of responsibility of a community living in the way of the Gospel seems a much more secure way to ensure that leadership cannot do evil in the hearts of the brothers. It may not be the most efficient way when there is a task to accomplish, but it is the safest way to ensure that everyone is well cared for while also doing his or her share in caring for the others, and it seems that St. Francis sees this as the best way into the Kingdom of God.

However, there are limits to this approach of understanding authority when an organization grows. While the 1221 Rule was the informal rule that grew with the order but was never approved, the 1223 Rule became the official rule of the Franciscans that has remained valid until today.[25] It is not dramatically different, and it retains the spirit of Franciscan simplicity and poverty that is characteristic of the 1221 Rule, but it lacks the many details that show the meaning of fraternity that we find in the 1221 Rule. The question of the use of authority is much more streamlined in the 1223 Rule, as the Franciscan Order was already quite large and would eventually grow even larger. It could no longer function as a

fraternity in the strict sense, or a community in which most brothers know the needs of most other brothers. Fraternity is not a realistic means of building functional large-scale political structures, but the experience of fraternal life does form competent leaders who may then be entrusted with hierarchical authority. For the concept of fraternity to be consistently applied in daily life, it must remain within what one person can experience as community. However, fraternity provides us with a hermeneutical key towards understanding relationships, including relationships of authority that extend beyond the individually experienced community. Furthermore, it lets us understand one person's autonomy as not just carving out a space for detached private decision making but as understanding human autonomy as a means to build up the interdependency of care that is at the foundation of human being.

### 3.3. Fraternity in the Canticle of Creatures

Another text to be considered here is the Canticle of Creatures, where St. Francis speaks together with all creation as a fraternity that forms a family in praise of God.[26] He composed this canticle shortly before his death, and he composed it not in Latin but in his Umbrian Italian dialect, which is his native tongue and the language he would use in speaking with a friend. He addresses God as "mi signore," uncharacteristically for him using the personal possessive pronoun in the first person singular, which further emphasizes the closeness with God that he felt while composing this canticle. In the first two verses, St. Francis sings on behalf of all humanity the praises of God to whom alone all praise is due. But St. Francis also recognizes man's unworthiness before God, on account of man's fall into separation from God. The canticle continues with verses 3 to 9, in which St. Francis sings of how all creatures praise God. But these are not the creatures of our world, akin to the praise in the Canticle of the Three Young men in the Book of Daniel. Not the animals and plants or even ourselves give praise, but the heavenly creatures of sun, moon, and stars, and the earthly creatures of wind, water, fire, and earth. It is the perfection of the heavenly creatures and the perfection of the original earthly elements of which all is made that praises God. But in this Canticle, he addresses each as brother or sister, applying the concept known to us for individual real beings rather than abstractions. Fraternity is the concept that bridges the experienced reality of actual creatures in their imperfections and idealized creatures in the perfection of God's creation.

Sir Brother Sun[27] at the start and Sister Mother Earth at the end frame the praises of the creatures. There is a family formed by these creatures, and while God is their origin, the images of parental care are retained, now both male and female, in an overall order of creatures and creation that is fundamentally fraternal. In the remaining four verses, the canticle speaks of human reconciliation, human suffering, and the specifically human experience of knowing the coming of our own death. In this part, the Canticle returns to the human being who through being a peacemaker is giving praise to God. Through reconciliation among each other, we reconcile the world with God to whom we hope to return in God. Working towards reconciliation in awareness of our death and desiring return to God is our place in the fraternity of creatures.

St. Francis's companions left us with a description of how the canticle was composed.[28] The first nine verses respond to a vision that he had while he was in great suffering, and this vision promised him not only eternal life in heaven but also assured him that he was already partaking in this life now. This makes these verses an eschatological vision, or a vision of the world to come as it is restored to full harmony with God at the end of time. He later amended the Canticle by adding the last four verses that speak of human reconciliation and the coming of death. His companions would write that this was done in the context of a conflict in his hometown that threatened the peace that St. Francis wanted to preserve. Jacques Dalarun insightfully suggests reading the Canticle as a drama in three parts that begins, in the first two verses, with the experience of separation from God. It continues up to the end of the 9th verse by singing of the goodness of God's creation, and it concludes in the last part by calling for reconciliation so that a peaceful death may return

us to God (Dalarun 2015). This dramatic reading of the Canticle considers it a call to action that leads to concrete consequences of our understanding of nature as creation.

Therefore, the Canticle of Creature challenges us to apply the concept of fraternity not only to communal life of individuals who already share much common ground in their outlook on life but also to understanding concretely embodied natural being when it is not like us. Fraternity with creatures lets us interpret the embodied biological life of all creatures, including ourselves. When fraternity is understood out of its original meaning of communality of biological interdependency of embodied being, then it unlocks the meaning of our relationship to nature, including our own nature.[29] It interprets embodied weaknesses, defects, needs, and dependences not negatively as mere limitations to our human autonomy but positively as a call to respond through life-giving community.

We must now return to the remarkable absence of animals and human beings from the creatures that give praise in (following the divisions made by Dalarun) the second part of the Canticle, or verses 3 to 9. However, the earthly elements are not merely idealized and abstract but described as useful, by producing the herbs and fruits that sustain us and the animals. The being of creatures is understood as caring for others. But each creature is also valued for what it is in itself. It is not as if each creature were valued only in as much as it is useful, but its inherent value is seen also in its utility. Indeed, in the understanding of fraternity, we value a brother or sister not only for the security of fraternal support for us. Instead, we recognize a brother or sister as one equal to me whose life I value as I value my own. Paraphrasing Scripture, we could say that we love our brother and sister as ourselves.

This value is not merely the sum of any concretely present momentary powers. In recognizing a brother or sister, we recognize a past and a present, and we already anticipate a future that is implied by the past and present. A life is always such a story. In a brother or sister, we recognize the other as a story and not just a current fact whose subsequent future is without basis in real being and of no concern to us now. The potential of the future is embodied in concrete biological being, rather than merely in our imagination, in the already realized past and in the present reality with powers and needs. Sustaining the reality of this future becomes a duty on those who are called to sustain it by recognizing the other as brother or sister.

## 4. Applications

The fraternity that St. Francis sees between us and all other creatures on account of the common nature of their createdness out of the four worldly elements remains relevant. Today, we understand our own creaturely nature not through the elements of classical understanding of nature, but through the concepts of molecular biology. All living beings share certain fundamental biochemical pathways, such as the generation of ATP through glycolysis or, if dependent on oxygen, the citric acid cycle. All life on earth depends on DNA and DNA-replicating and -transcribing proteins. Living beings share a common origin and ancestry in the evolution of life on earth. The sense of fraternity with creatures has a real foundation in our knowledge of biology. Our fraternity with them lets us recognize their inherent worth, their needs, and their abilities. In fraternity, we understand the need to respect them.

What has been accomplished by the approach of considering fraternal relationships with natural beings is to see the individual living being not merely as it is in its status quo, not merely as a material system at a moment in time, but as a living being that we can understand as if a story told in time, a life story, that has begun and implies a future. It has a potential that is already real, and this potential, not only a living being's current state, is relevant for my ethical stance towards it. What has been accomplished by the introduction of the theme of fraternity is that the ethical discernment is no longer distinct from biological knowledge, but it integrates it into the larger context of fraternal relationships. Placed in this context, killing a human embryo is understood as an ethically different matter from killing a chimpanzee embryo, even if the human embryo differs from a chimpanzee embryo in nothing more than a few percent of its DNA sequence and in none of the concrete features

of either embryo. Understanding nature through fraternal relationships lets us see the different obligations that we have to different creatures on account of the potential of their lives or the continuation of their story. Fraternity thus understood does not deny difference. Human beings stand out from other creatures by their sense of responsibility, such as the ability to reflect on one's actions and consider their ethical value. This qualitative difference can be called upon to justify the special ethical concern that we have towards other human beings. It is only human beings that will, if usually only for part of their lives, recognize fraternity as an ethical value, rather than just a biological fact. Fraternity lets us discern another creature's life as a story that ought to be told.

Understanding our being as fraternity in nature also lets us understand the value of autonomy. We speak of it as human freedom. We can make decisions that we consider good, and while we do not always agree on individual perceptions of the good, we can agree on the importance of a framework that respects human autonomy. We can attempt to build such a framework in which dialogue between autonomous actors might lead to consensus.[30] But what would be the common ground that lets diametrically opposed interests reconcile in compromise? Instead of seeking common ground and a matching framework for reconciling interests in political theory, we may consider the one that we find in the understanding of nature. There, we find that our autonomy is brought about through a process of nurturing and education. Our autonomy is partly grounded in our biological nature that makes autonomy possible, but it is also grounded in the ethical actions of others who supported us enough to let us develop our capacity to exercise this autonomy. The gift thus received obliges us to the giver of the gift, and here, it is the community. Calling for respect for unborn human life, on account of the embodied human being with its own potential, becomes a legitimate interest for which to advocate.

For the same reasons, our mutual dependency and mutual vulnerability through infectious disease justifies collective decision making in vaccination. When knowledge of nature establishes that a specific vaccine carries a lesser risk than the illness from which it protects, or even no risk at all, then restricting individual autonomy by mandating vaccines follows from our fraternal interdependence.

In conclusion, what has been accomplished is to show how fraternity gives human autonomy a sense of direction that now can include knowledge from biology as a rationale for ethical acting. It is not a naturalism that idealizes a certain concept of nature, as if what happened in the absence of conscious and ethically informed human acting were normative for us. Instead, it understands human nature as an embodied nature that exists in fraternity with others. Neither does it unduly privilege our human relatives in ethical decision making, as if the needs of one who is my biological brother must always be more important to me than the needs of one who is not. Fraternity is a relationship of mutual understanding of dependency on care, and through fraternity, I understand the needs of individual creatures and the meaning of my autonomy. It brings together what would otherwise by irreconcilably opposed—the interests of different people with different conceptions of their own personal good. Our shared human nature and the shared nature of living beings is the best foundation on which to find common ground in times of political polarization and environmental crisis.

**Funding:** This research received no external funding.

**Conflicts of Interest:** The author declares no conflict of interest.

## Notes

1　(General Curia O.F.M. 2004).

2　*FA:ED* I: 113–114 (https://www.franciscantradition.org/francis-of-assisi-early-documents/the-saint/writings-of-francis/the-canticle-of-the-creatures/129-fa-ed-1-page-113).

3　https://ourworldindata.org/famine-mortality-over-the-long-run (accessed on 20 August 2022).

4　https://ourworldindata.org/world-population-growth#how-has-world-population-growth-changed-over-time (accessed on 20 August 2022).

5    This is also an important theme in the encyclical Laudato si', https://www.vatican.va/content/francesco/en/encyclicals/documents/papa-francesco_20150524_enciclica-laudato-si.html (accessed on 20 August 2022). par. 101–5.

6    https://www.supremecourt.gov/opinions/21pdf/19-1392_6j37.pdf (accessed on 20 August 2022).

7    See note 6, Dobbs v. Jackson, p. 38.

8    Of course, biology cannot pinpoint the exact moment when a new living being begins or define life and its meaning. But by the time abortion is considered, the presence of a distinct living being is unambiguous in biology. It is what is studied in embryology or meant to be removed by performing an abortion.

9    https://www.bundesverfassungsgericht.de/e/fs19930528_2bvf000290en.html (accessed on 20 August 2022).

10    Die Würde des Menschen ist unantastbar. Sie zu achten und zu schützen ist Verpflichtung aller staatlichen Gewalt.

11    Jeder hat das Recht auf Leben und körperliche Unversehrtheit. Die Freiheit der Person ist unverletzlich. In diese Rechte darf nur auf Grund eines Gesetzes eingegriffen werden.

12    https://www.cdc.gov/coronavirus/2019-ncov/vaccines/safety/safety-of-vaccines.html (accessed on 20 August 2022) and (Belongia and Naleway 2003). https://www.ncbi.nlm.nih.gov/pmc/articles/PMC1069029/ (accessed on 20 August 2022).

13    See, for example https://www.webmd.com/lung/news/20210729/fighting-fauci-from-ridicule-to-death-threats-attacks-continue. (accessed on 20 August 2022).

14    https://www.un.org/en/about-us/universal-declaration-of-human-rights (accessed on 20 August 2022).

15    https://archive.osb.org/rb/text/toc.html (accessed on 20 August 2022).

16    Rule of St. Benedict, Chapter 6. Of course, Benedictine life in practice places a strong emphasis on liturgy and communal prayer. Here, however, my emphasis is on the rule and what the rule says about the life of the community.

17    Rule of St. Benedict, Prologue.

18    ER, Chapter 1 (*FA:ED I*, 63–64).

19    *FA:ED I*, 80–81.

20    This is not to suggest that hospitality is not of the greatest importance to the monastic community. The visitor is meant to be received like Christ (chp. 53).

21    Rule of St. Benedict, chp. 2.

22    Except for two quotations from scripture, Lk 14:26 and Mt 23:8, in which Christ is the teacher.

23    *FA:ED I*, 61–62.

24    For example, consider Jean Vanier, the highly respected and much-admired founder of l'Arche, whose abuse of vulnerable adults did not come to light until after his death.

25    *FA:ED I*: 99–106.

26    *FA:ED I*, 113–114. For a detailed study of the Canticle see (Hammond 2011).

27    In the Umbrian text, God is referred as *signore*, whereas the son is referred to as *messor lo frate sole*. Again, fatherhood is not explicitly mentioned, but it now seems implied by the context.

28    AC 83–84 (*FA:ED II*: 186–187).

29    This connection between understanding nature and the dignity of unborn human life is also noted in the encyclical *Laudato si'*, which takes its title from the Canticle of Creatures. In paragraph 120, it says "Since everything is interrelated, concern for the protection of nature is also incompatible with the justification of abortion".

30    For example, consider John Rawls's "veil of ignorance" to structure such dialogue. It respects the individual decision maker in deliberations and does not subordinate the individual to the collective as in utilitarianism. But it brackets the individual's peculiar social place with its privileges and asks each to consider their decision making as if they did not know their individual place in the social order. But since no social experience exists in which individuals are unaware of their place in their social order, what Rawls suggests may be impossible. The order of nature that is open to investigation seems the more secure ground for consensus apart from already formed convictions about the social order.

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
