# Peer review of "Fraternity as Natural Being"

_religions, doi:10.3390/rel13090812_

Round 1
Reviewer 1 Report
This seems a reasonable attempt at demonstrating how, at least theoretically, religion can actually play a good and useful role here—here in this polarized society. In attempting not to agree or disagree with the author(s) ideas per se, here are a few thoughts to consider in terms, rather, of presentation and clarity.
•to make the argument clearer, please define up-front briefly what exactly is meant by ‘natural being’, especially since this is so key to the story.
•fraternity, the concept, as a (potential) ‘in-between’ or a reconciler of autonomy and natural being is a good and strong idea. The problem is fraternity with WHOM. It reminds me of oxytocin, the hormone/pheromone that is involved in love between mom and baby, between lovers, but also seems to ‘favor’ love of others like you (https://www.ncbi.nlm.nih.gov/pmc/articles/PMC3029708/).
—Consider addressing that most if not all major religions make the central point of St. Francis discussed here—that all men and women are brothers, and interdependent (admittedly some require certain conditions are first met, like you must believe in me or him or this or that) and some even claim, as does Francis, that we are interdependent with all creatures and the universe, and yet we are still superbly polarized. Can religion/Franciscans actively do anything with the concept of fraternity outlined here?
•Vaccine mandates as an example makes sense in this context, but I’m not sure about abortion; climate change, for one, might be better. Abortion has some tough problems made evident even in the authors own arguments; for example, after saying humans have the same DNA and basic biochemistry as all organisms (lines 389-400), then says a human embryo and a chimpanzee are different, which seems exactly the opposite conclusion to draw from the previous argument (in addition, it’s not entirely clear that other primates, whales, elephants, dolphins, etc don’t indeed have and ethics or ethical behaviors). Similarly, the ‘potential life’ discussion is unconvincing here, as embryos themselves, how they are defined, as alive or not, is not agreed upon by anyone—scientists or theologians amongst or between themselves, even.
•line 379: four earthly elements kind of appears out of the blue here. Referring to the ancient concepts of fire, water, etc?
minor grammatical or related points:
Line 22 and 152: strive should be strife
120-121: language unclear, speed and offered limited the
202-203: unclear, more insightful than it for those wanting
231: supposed should be supposedly
283: add ‘for’ after cared
Reviewer 2 Report
review Article Fraternity as Natural Being
Novelty: The theme of the article and its main focus, natural being, are original. The question (fraternity thought as natural being as an antidote to political division) is clearly stated at the beginning. The article provides new knowledge with a fresh new perspective. (The problem, however, is the form of the argumentation, see below.)
Scope: The work fits the journal and special issue scope.
Significance: This article is more an essay consisting of a stream of personal argumentation than a discussion with the literature. There are no “results to be interpreted”. The argumentation is in a sense original and significant, but is not connected to existing scholarly debates. The two social issues discussed, abortion and vaccination, have only illustrative value but do not contribute to the social or scholarly debate on these issues.The conclusions are thus not supported by results, unless purely internal, philosophical ones. Quality: The article is inappropriate as a contribution to existing debate. There is almost no embedding within current discourse. Take, for example, the meaning of fraternity, where no other literature is mentioned (155). Parts of the argumentation are dense and incomprehensible (e.g. 352-353, 361, 368-375, 405, 415-417). The highest standards for presentation would include more referencing and a clearer return to the main question in the conclusion. The main problem: the author discusses the importance of natural being through several stages (introduction, two cases, fraternity, rule 1221, and canticle) whose connection remains arbitrary and incoherent. The importance for “political healing” - the main question - of the chosen themes (fraternity, natural being or the rule of 1221) as compared to alternatives (e.g.. humility, law, the rule of 1223) remains unconvincing. The conclusion, furthermore, introduces new elements not dealt with before (see below).
Scientific Soundness: The design is correct, but the substantiation of the argument is weak and one-sided. There is nothing wrong with the analyses, e.g. those of abortion or the Rule of 1221, but the connection between the analyses in the paragraphs restst purely on internal reasoning, not external evidence. The description of the themes rests almost completely on individual interpretation; references to the latest literature are extremely scarce.
Interest to the Readers: The conclusions may be interesting for the readership of the journal. Due to its interdisciplenary approach (natural sciences and ethics and spirituality) it may attract a wide readership.
Overall Merit: The overal benefit to publishing this work would be the interdisciplinary approach mentioned above: the connection between fraternity and natural being and the modern cases combined with the old sources is original. The elaboration of the connection, however, remains thin and stricty philosophical. I am not convinced that the author has shown how and why fraternity (due to its natural beingness) would offer concrete solutions to the matter of political division. I mean with the power of argumentation like in pope Francis’s Fratelli tutti, for example.
English Level: The English language is appropriate and understandable. There are a couple of mistakes.
The weakest part of the article is the introduction, in which many unsupported or untrue statements are made:
+ “lives damaged by political strive”: how so?
+ “celestial bodies and the earthly elements as brothers and sisters”: there is no opposition heaven-earth in the Canticle.
+ “It is also the age of science and its triumph over two of the three traditional banes of human existence—pestilence and hunger”: Think of Bacon and universities and the 13th century can be called the age of science. And have the pest (corona) and hunger really been overcome in modernity?
+ “How could such an increase of power in human hands not lead to irreconcilable conflicts between different factions and their different conceptions of the good and their hopes for their future?”: this is already very medieval.
+ “dependence on nature”: why is this dependence necessary for a shared awareness of the good?
+ “Autonomy from nature”: How autonomous are we moderns, considering corona and climate change?
+ “it would be mastered by those who are powerful”: in what sense?
+ “for political polarization that now seems to emerge as a common theme in the societies of the “West””: substantiate with current discussions and data. Read Jonathan Haidt on the righteous mind, for example. One cannot deal with this while neglecting the debate.
+ “or the society in which the scientific-technological revolution with its mastery over nature began”: this is already very medieval.
+ “knowledge of nature conflicts with human autonomy”: how so? And why should human autonomy not be supported knowledge of nature??
Furthermore, after the introduction:
+ transformed (180): but it is already both (155-156)
+ “Monasticism remains fundamentally defined by the solitary life of each monk even when it is lived as..” (195): This is incorrect: think of the role of the liturgy. And is this not a contradiction?
+ “until the 58th chapter” (223): this is because the liturgy comes first.
+ “Here, the newcomer is not meant to experience the monastic community as a welcoming fraternity” (226): what is the source of this strange statement?
+ “maternal care” (269): this is already a monastic motive (e.g. cistercians).
+ “community living in the way of the Gospel seems” (280): decisive is the mobile way of life.
+ “but as a story, a life story, that has begun and implies a future”: This has not been explained enough. How does fraternity relate to story? What is the source? And the connection with narrative theory? Why story here and not earlier?
+ note 30: Also consider the massive criticism given to Rawls, if you decide to mention him.
+ “Instead of seeking such a framework in political theory” (412): nobody does that.
+ education (414)/ gift (417): how is education nature? How is a gift nature?
+ “a legitimate interest for which to advocate” (419): This political position of the author (“pro-life as the best option”) seems to undermine exactly the author’s own argumentation above and to reinforce the political division. Does fraternity as natural being exclude those who are pro-abortion?
+ “follows from our fraternal interdependence” (425): This has not been explained enough. Why fraternal? Why is not legal or ethical enough of an argument?
+ “that now can include knowledge from biology as” (427): the problem that fraternity is NOT biological in the religious life has not been solved.
+ “through fraternity, I understand the needs of individual creatures” (434): why do we need fraternity for that? The argumentation above has not convinced me that I need brothers in order to understand those needs.
+ “their own personal good” (437): this is a new element. The author’s discussion was about conceptions of the COMMON good.
+ “to find common ground” (438): is this the same as healing (cf. the main question)?
Round 2
Reviewer 2 Report
the paper remains interesting and inspiring, but the main obstacle to publication in a SCHOLARLY volume (or theme issue) has not been resolved: there is almost no scholarly contextualization, which means that the paper is more a philosophical essay that a thorough analysis/presentation of theological research results. There is almost no engagement with other scholarship.